# Targeting the Limbic System: Insights into Its Involvement in Tinnitus

**DOI:** 10.3390/ijms24129889

**Published:** 2023-06-08

**Authors:** Anurag Singh, Paul F. Smith, Yiwen Zheng

**Affiliations:** 1Department of Pharmacology and Toxicology, School of Biomedical Sciences, University of Otago, Dunedin 9016, New Zealand; paul.smith@otago.ac.nz; 2Brain Health Research Centre, University of Otago, Dunedin 9016, New Zealand; 3The Eisdell Moore Centre for Research in Hearing and Balance Disorders, University of Auckland, Auckland 1023, New Zealand

**Keywords:** tinnitus, auditory system, limbic system, synaptic plasticity, hippocampus, amygdala, pharmacology

## Abstract

Tinnitus is originally derived from the Latin verb *tinnire*, which means “to ring”. Tinnitus, a complex disorder, is a result of sentient cognizance of a sound in the absence of an external auditory stimulus. It is reported in children, adults, and older populations. Patients suffering from tinnitus often present with hearing loss, anxiety, depression, and sleep disruption in addition to a hissing and ringing in the ear. Surgical interventions and many other forms of treatment have been only partially effective due to heterogeneity in tinnitus patients and a lack of understanding of the mechanisms of tinnitus. Although researchers across the globe have made significant progress in understanding the underlying mechanisms of tinnitus over the past few decades, tinnitus is still deemed to be a scientific enigma. This review summarises the role of the limbic system in tinnitus development and provides insight into the development of potential target-specific tinnitus therapies.

## 1. Introduction

Tinnitus and tinnitus disorder are the two main classifications used to describe tinnitus and tinnitus-related distress. While tinnitus is the conscious perception of a tone or composite noise that has no corresponding external auditory source, tinnitus disorder can include cognitive impairment, emotional discomfort and functional disability [1]. Tinnitus is a frequent auditory disorder associated with various disease conditions (both otological and non-otological) [2] and affects millions of people worldwide. Over 70 million people in Europe and more than 50 million in the United States are affected by tinnitus [2,3,4]. At least 0.5 million Australians [5] and 0.2 million New Zealanders (>14 years old) have experienced audible tinnitus at some stage during their lives [6]. The common risk factors for tinnitus are noise exposure [7,8], Meniere’s disease, head injury or other trauma, ear infection, lifestyle issues, psychological problems and ageing [9,10] (Figure 1). Due to the inclusion of people with self-reported tinnitus and high phenotypic variation in large genomic studies, the extent of genetic involvement in severe tinnitus is still unknown [11]. However recent studies have identified rare genes that are associated with severe tinnitus [12,13]. Approximately 10% of patients suffering from tinnitus develop mild or severe impairment. Severe tinnitus is strongly linked to anxiety, sleeplessness, difficulties concentrating, poor psychological well-being and poor quality of life [14,15]. Consequently, tinnitus is becoming more of a burden on healthcare systems [16] and produces a negative impact on economic growth due to a compromised workforce [17]. Moreover, tinnitus patients worldwide try a range of possible treatment options, hence the healthcare cost involved is significant [18,19]. While some medications, such as certain antidepressants [20,21] and anticonvulsants [22,23], may help reduce the severity of tinnitus symptoms, the lack of an FDA (Food and Drug Administration, USA)-approved drug indicates that none of the drugs tested completely abolished tinnitus. In addition to drugs, an increasing number of both invasive and non-invasive techniques, such as cochlear implants [24,25], removal of the vestibulo-cochlear nerve [26], surgical decompression of the auditory nerve from the compressed veins [27], transcranial magnetic stimulation (TMS) [28,29], transcranial electrical stimulation (TES) [30], bi-modal stimulation [31,32] and others [33,34,35], have been used in the treatment of tinnitus. However, there is no sufficient evidence supporting the effectiveness of these interventions. Tinnitus disorder highlights the condition’s all-encompassing character and its profound effects on people’s well-being.

It is well recognised that tinnitus may be generated from cochlear damage that causes aberrant activity in the cochlear nerve [36] at the periphery of the auditory system, which in turn causes hyperactivity in the dorsal cochlear nucleus (DCN) [37,38], inferior colliculus [39,40] and auditory cortex [41,42], as well as tonotopic map reorganization in the central auditory system [42,43,44,45]. Although the mechanism for this is not fully investigated, a possible explanation could be an alteration in the balance of excitatory and inhibitory neurotransmission in the auditory and associated networks. A study consisting of 10 tinnitus patients suggested that the alterations in the degree of neuronal activity in the DCN are correlated with modifications in tinnitus perception [46]. In animal behavioural experiments, a significant association between the peak level of DCN activity and the behavioural measures of tinnitus was discovered [47]. Moreover, the long-term effects of acoustic over-exposure resulted in increased burst firing and synchrony in the inferior colliculus [48]. Noise trauma has been shown to enhance spontaneous firing rates (SFRs) and bursting activity in the auditory cortex in a variety of animal models of tinnitus, in both acute [41] and chronic conditions [49]. Therefore, the contribution of neuronal hyperactivity in the auditory pathways to the development of tinnitus has been well-accepted [36,44,50].

## 2. Tinnitus: Theoretical Models and Role of the Limbic System

### 2.1. Current Theoretical Models of Tinnitus

Traditionally, tinnitus was considered to be a pure otological disorder, but modern theories have challenged this and proposed that tinnitus is a group of illnesses with multiple physiological mechanisms.

The most common attribute of tinnitus is a constant phantom sound (either a pure tone, hissing or roaring) in the ears in the absence of the corresponding external auditory stimulus [51,52]. This phantom sound is more noticeable in silent environments and hence an array of studies has compared the brain activity in tinnitus patients and healthy controls in silent environments to investigate the abnormal brain activity linked to tinnitus using electrophysiological (e.g., EEG) or magnetoencephalographic (MEG) recordings. Abnormal spontaneous activity, such as elevated gamma oscillations [53,54,55,56] and reduced alpha [51,57,58] activity, has been reported in the auditory brain regions of tinnitus patients. Interestingly, a spectral examination of the brain activity in a salicylate-induced animal model of tinnitus also showed a significant decrease in alpha and an increase in gamma band activity in the auditory cortex [59], which is consistent with neuromagnetic recordings in humans with acute tinnitus, reported by Lorenz et al., 2009 [60]. Therefore, it was generally thought that synchronised gamma-band activity links sensory events into a single cohesive conscious perception [61], and the ongoing gamma-band activity in the auditory cortex is necessary for tinnitus to occur [54]. The theoretical foundation of these observations forms the basis of the thalamocortical dysrhythmia (TCD) model [62] and the synchronization by loss-of-inhibition model (SLIM) of tinnitus [63]. The TCD model identifies the establishment of increased spontaneous firing of thalamic fibres as a crucial component of the development of tinnitus [62]. In particular, due to the reduced excitatory sensory stimuli from the damaged inner ear to thalamic relay cells, the hyperpolarised cell membrane induces the relay cells to fire low-threshold calcium spike bursts in a slow-wave form [62]. The formation of this slow-wave rhythm in the cortical neurons is subsequently brought on by thalamocortical feedback loops, which are detectable as continuous delta activity on the scalp. Tinnitus patients exhibit a decrease in delta activity in the cortical area [64]. The SLIM model hypothesises that the rise in the gamma frequency range may potentially be caused by a reduction in lateral inhibition processes in the auditory cortex as a result of the reduced activation of inhibitory neurons such as those which are GABAergic [63]. Increased gamma activity could also be due to hearing loss-related reduction in sensory input, which in turn reduces alpha-mediated inhibition. This leads to heightened gamma activity, as alpha band activity ordinarily suppresses gamma activity and hence results in increased neuronal synchrony, which is thought to play a role in the cause of tinnitus [52]. Thus, this discrepancy between cortical suppression and excitation offers a theoretical explanation for the alpha-up, delta-down pattern commonly observed in the resting-state non-invasive magneto- and electroencephalography (M/EEG) data of tinnitus patients [65]. In summary, based on the TCD and SLIM models, hearing loss may cause tinnitus by interfering with the coherent oscillatory activity between the thalamus and cortex. Although the mechanism which connects hearing loss with tinnitus remains unclear, both are well-known and mutually related medical conditions [66,67].

In addition to the above, reduced alpha activity is correlated with a desynchronised neuronal network that is often linked with auditory attention [68,69]. However, the source of alpha activity and whether it is induced by inhibitory activity or some other network-generated factors, remains elusive. Given that only 10% of synchronously activated neurons can generate an amplitude that is around 10-fold greater than that of unsynchronized neurons [70], and that approximately 10–15% of cortical neurons are GABAergic [71], EEG alpha activity, in theory, could be due to periodic fluctuations in the activity of inhibitory neurons. Therefore, one can speculate that tinnitus may be associated with a problem in areas of the brain where irrelevant incoming information from sensory regions of the human brain is actively suppressed under normal conditions [72,73], i.e., the “noise cancellation” model.

As proposed by Rauschecker, et al. [74], in a non-tinnitus case, unwanted repetitive auditory information is cancelled out at the level of the thalamus (medial geniculate nucleus; MGN), whereas in tinnitus, the cancellation mechanism i.e., the auditory gating mechanism, is compromised, which causes the MGN to become uninhibited, and eventually results in the perception of tinnitus sound [74,75]. Anatomically, the auditory gating system is thought to involve the auditory cortex, thalamus, prefrontal cortex, and limbic and paralimbic areas. The thalamus is innervated by the serotonergic fibres originating from limbic and paralimbic structures [76,77] and plays a role in auditory-limbic interactions. It has been speculated that at least certain limbic areas work as a component of a feedback circuit to the auditory system that plays an “inhibitory gating role” for auditory perception [78]. To predictably adjust and enhance auditory performance, auditory gating may also act as an adaptive process to filter out unnecessary event-based or temporal information [79]. Thus, the limbic system determines which noises are unimportant and blocks them before they draw our attention. However, in the case of tinnitus, it is hypothesised that this gatekeeping function may be weakened so that the tinnitus sound is allowed to reach the auditory cortex for conscious perception [74].

Indeed, mounting pieces of evidence over the years have suggested that tinnitus is associated with functional and structural alterations in brain regions involved in emotional regulation, presumably as a result of consciously or subconsciously evaluating the continuous noise negatively, and failing to acclimate to it [80]. Functional imaging studies in tinnitus-affected humans and animals demonstrated activation of regions, both inside and outside the auditory systems. To explain this, one of the proposed hypotheses is that the abnormal filtering of auditory information by the limbic regions may play an exclusive role in tinnitus generation/perception. In the late 1980s, a study proposed a cross-talk between tinnitus and the emotional state of patients by showing that treating the comorbid major depression may lessen tinnitus disabilities [81]. It was later recognised by other groups that the limbic system may be actively involved in modulating or perpetuating tinnitus [36,82,83]. For example, the limbic and primary auditory areas showed dynamic linkages between tinnitus-related abnormalities, highlighting the significance of auditory-limbic interactions in tinnitus [84]. Functionally, the limbic system is an area of the brain that controls learning, memory development and storage, and emotions [85]. The hippocampus, amygdala, basal ganglia, cingulate cortex, and subcallosal area, among other limbic structures, are thought to contribute to a generalised “distress circuit” that can be triggered by real or phantom stimuli related to auditory, nociceptive, or other sensory stimuli [86,87].

### 2.2. The Hippocampus and Tinnitus

The hippocampus is situated medio-temporally in each hemisphere of the brain, and its role in mediating spatial memory and memory consolidation is substantially investigated and widely acknowledged [88,89,90,91]. The hippocampus receives auditory information primarily through the entorhinal cortex and is responsible for the temporal processing of the information [92,93]. The hippocampus also projects directly from area CA1 to the auditory association cortex and even to the primary auditory cortex [94] and plays a role in language, music processing and the development of long-term auditory memories [95,96]. For example, auditory recognition tests show very poor results in patients with extensive bilateral hippocampal loss [97], and research on monkeys suggests that auditory cues may be involved in spatial memory formation mediated by the hippocampus [98]. Using resting-state functional Magnetic Resonance Imaging (MRI), several studies have reported structural alterations such as significant loss of grey matter in the hippocampus of noise-induced tinnitus patients [99]. In addition, hippocampal activity and cognitive changes in human patients were found to be linked with the unpleasantness of tinnitus [100,101]. These findings suggest anatomical and functional connections between the hippocampus and tinnitus perception and/or tinnitus-related distress. Genetic factors have recently been associated with severe tinnitus, such as an increased occurrence of rare variants in ANK2 and TSC2 synaptic genes, which exhibit high expression in the hippocampus and cortex [12]. Furthermore, one study reported an increased occurrence of specific genetic changes such as missense and large structural variants in regions of the genome that are highly conserved. They identified CACNA1E, NAV2 and TMEM132D as potential genes that may play a role in contributing to severe tinnitus [13]. However, additional research is necessary to establish a definitive genetic linkage between tinnitus and the hippocampus.

Interestingly, noise exposure and listening to music have also been shown to cause structural and molecular changes in the hippocampus. For example, noise-exposed animals exhibited elevated mitochondrial areas inside the hippocampal neurons [102], possibly due to the variations in synaptic transmission that trigger presynaptic mitochondria’s ultrastructural plasticity to respond to the metabolic demand. Noise-exposed animals also exhibited a significant reduction in the optical density of Nissl bodies [103], a reduced postsynaptic density and outspread synaptic clefts [104] in the hippocampus, indicative of impaired synaptic transmission and neural activity.

Overall, these studies suggest that hearing loss produces a negative impact on hippocampal anatomy and function (Figure 2). However, the link between tinnitus and hippocampal function, especially the relationship between tinnitus and hippocampal synaptic plasticity, remains to be elucidated.

### 2.3. The Basal Ganglia and Tinnitus

The basal ganglia consist of a group of subcortical nuclei, such as the striatum, subthalamic nuclei, substantia nigra and globus pallidus. They are situated at the top of the midbrain and the base of the forebrain and are primarily involved in controlling voluntary movements such as eye movements, aiding in balance, and supporting posture [105]. In the last two decades, the striatum (a nucleus present in the basal ganglia, critical for motor and reward systems) was found to not only mediate sensory information transmission to the cerebral cortex but may also play a role in tinnitus [106,107]. Studies in monkey [108,109], cat [110], rat [111] and human [112] demonstrate a functional and anatomical link between the caudate nucleus and auditory cortex. A case report of tinnitus cessation following a cerebrovascular accident that lesioned both the caudate nucleus and putamen (dorsal striatum) revealed the clinical viability of striatal neuromodulation to reduce tinnitus [113]. Over the years, a few studies have reported that deep brain stimulation (DBS) of the striatum can significantly suppress symptoms in patients suffering from tinnitus [114,115,116]. DBS of the striatum has also been tested in an animal model of tinnitus, where electrical stimulation of the caudate nucleus attenuated cluster neuronal firing in the auditory cortex and suppressed tinnitus-like behaviour [117]. Overall, the caudate nucleus may be involved in tinnitus; however, the mechanisms through which it regulates the disorder remains poorly understood.

The basal ganglia also have a limbic sector which consists of the ventral pallidum, nucleus accumbens (NAcc) and ventral tegmental area (VTA) [118,119]. The NAcc is strongly associated with addiction and depression [120,121,122]. The projection from the anterior cingulate cortex to the NAcc-VTA region has been found to mediate the effects of unpleasant sound [123], music [124], as well as tinnitus [125]. According to high-resolution magnetic resonance imaging studies in the brain, tinnitus patients exhibit structural and functional abnormalities such as a decrease of gray matter in the NAcc [87,125]. Recent research has also demonstrated that the severity of tinnitus and hyperacusis is linked to aberrant neuronal excitability in the NAcc, which results in emotional alterations to a sound stimulus [126,127,128]. Therefore, the NAcc has been suggested to be implicated in tinnitus and may contribute to changes in the limbic-auditory connections [129]. However, it remains unclear how the NAcc might be involved in the perception of tinnitus.

One critical structure of the basal ganglia, the subthalamic nucleus (STN), which is not directly linked to the auditory system, but is connected to the NAcc, has also been found to play a role in tinnitus [130]. For example, DBS in the STN of a tinnitus patient significantly improved tinnitus handicap inventory (THI) scores compared to the situation prior to DBS [115], suggesting that DBS of the STN may have a beneficial effect in the treatment of tinnitus. All of this evidence indicates the possibility that the STN contributes to tinnitus mechanisms, however, the exact nature of this connection still requires further investigation.

### 2.4. The Amygdala and Tinnitus

The amygdala is a key mediator of the emotional and other behavioural responses to sensory stimuli, across all senses, and is connected to the limbic, executive, and other sensory areas of the forebrain [131]. Anatomical investigations in different animal species have shown relatively consistent networks of projections to the nuclear complex of the amygdala from auditory and auditory-associated regions [110,132,133,134,135]. Importantly, there is a second pathway from the basolateral amygdala to the principal inhibitory nucleus in the thalamus, the thalamic reticular nucleus (TRN; Figure 3) [136]. The TRN consists of a layer of inhibitory GABAergic neurons present between the thalamus and neocortex and produces inhibition of the sensory thalamocortical relay neurons [137]. The cortex and thalamus simultaneously send excitatory collaterals to the TRN [138]. These connections put the TRN in the role of a gatekeeper, regulating the flow of sensory information from the thalamus to the cortex by evaluating sensory stimuli based on their behavioural relevance [139,140,141]. Using a computational model of thalamocortical circuitry, it was found that the stimulation of the basolateral amygdala inputs to the TRN results in decreased spontaneous thalamic activity [142]. Thus, the amygdala may play an important role in regulating auditory gating functions. Furthermore, resting-state fMRI investigations in tinnitus patients demonstrated an abnormal functional connection between the auditory cortex and the amygdala [143,144], which may be related to tinnitus-related distress given that the auditory cortex projection to the amygdala play a potential role in mediating auditory fear conditioning, and hence connects emotion with tinnitus perception [74,145]. Interestingly, a study reported that salicylate administration into the amygdala can significantly enhance sound-evoked local field potentials in the auditory cortex, changes indicating heightened perception and emotional salience of the tinnitus [146]. Thus, one can speculate that the functional connections between the “amygdala-TRN-auditory cortex” may play a potential role in auditory gating mechanisms, that may further contribute to tinnitus perception and tinnitus-related distress.

The significance of emotional memories is a major aspect of chronic tinnitus. Emotional memories contribute to chronic tinnitus by causing a sustained level of hypervigilance, thereby promoting a continuous level of awareness [147,148]. Sound-induced amygdala responses are found to be sensitive to their emotional strength [149] and align with the significance of sound in an individual’s sensory environment [150]. Overall, one possible theory could be that the amygdala may deliver a significant negative emotional signal to the auditory cortex, influencing the perception of acoustic information [151]. However, there is a lack of understanding of the mechanisms involved and investigating the role of the amygdala in mediating tinnitus induction or consolidation following noise exposure may offer a better understanding of the condition.

### 2.5. Other Limbic Regions and Tinnitus

The cingulate cortex is another part of the limbic system and is primarily involved in emotional responsivity [152,153], emotional processing and inhibitory control [154,155]. Interestingly, tinnitus discomfort has been associated with increased activity in the cingulate cortex [156,157]. Reportedly, manipulations that induce tinnitus resulted in increased fos-like immunoreactivity and Arc protein expression in the cingulate cortex of gerbils [158,159]. In contrast, one study reported no changes in the neuronal excitability or frequency response in the cingulate cortex after tinnitus induction with salicylate [160]. Furthermore, the subcallosal area (medial prefrontal cortex, orbitofrontal cortex, and anterior cingulate areas) is an important hub linking limbic-affective systems with thalamocortical perceptual systems. Using anatomical MRI [125], significant grey matter volume reductions in the subcallosal region were found in tinnitus patients when compared to the controls. The subcallosal areas, such as the ventromedial prefrontal cortex (vmPFC) and NAcc, demonstrated a crucial function in the long-term habituation to persistently unpleasant noises by sending feedback projections to the TRN, which in turn selectively inhibits the MGN regions corresponding to the unpleasant sound frequencies [161] (Figure 2 and Figure 3). Interestingly, a significant volume loss in the subcallosal area of tinnitus patients has been reported [99,125]. The engagement of the subcallosal region, for example, is also modified by pain anticipation and perception and responds to the unpleasant effects of discordant music to variable degrees [162].

Overall, the limbic system occupies a critical role in elucidating the underlying molecular causes of tinnitus, as tinnitus-related volume loss in the limbic regions such as the hippocampus, amygdala, and subcallosal area is often due to atrophy of neurons and glial cells, leading to impairment of synaptic plasticity mechanisms [163]. Thus, the limbic system is well placed to play a crucial role in mediating tinnitus sounds from being perceived and targeting and suppressing tinnitus signals at these subcortical levels before they reach the primary auditory cortex may open new horizons for tinnitus treatments.

## 3. The Limbic System and Synaptic Plasticity in Tinnitus Development

One critical but less investigated area is synaptic plasticity in the auditory and limbic areas during the development and consolidation of tinnitus. In the past two decades, it has been demonstrated through animal studies that tinnitus might be a pathology of synaptic plasticity in multiple brain areas, including the auditory and limbic systems [164,165] (Figure 2). Synaptic plasticity refers to activity-dependent changes in the strength or efficacy of synaptic transmission and is thought to play a key part in the brain’s ability to convert fleeting experiences into enduring memory traces [166]. In terms of auditory processing, synaptic plasticity can also be defined as a mechanism through which the neural activity created by an event, such as music, alters brain function by modifying cellular properties and synaptic transmission [167,168]. Synaptic plasticity is thought to play an important role in the early development of healthy brain circuitry and altered synaptic plasticity mechanisms are assumed to contribute to neuropsychiatric and other brain disorders. For example, *N*-methyl-d-aspartate receptors (NMDA) are well recognised for their role in mediating synaptic plasticity, such as long-term potentiation (LTP; a candidate mechanism for memory formation) in the hippocampus of the mammalian brain [169,170,171,172,173]. Altered NMDA receptor trafficking contributes to an impaired hippocampus and cognitive functions [174], and other psychotic illnesses such as schizophrenia [175]. Interestingly, the mammalian cochlea also expresses NMDA receptors which can be activated during the regrowth process after excitotoxic injury [176], indicating the existence of NMDA receptor-dependent synaptic plasticity in the peripheral auditory hair cells. In addition, NR2B subunit-targeted inhibition of cochlear NMDA receptor activity by Ifenprodil (an NR2B antagonist) was able to prevent noise-induced and salicylate-induced tinnitus in rats [165]. NMDA receptor-dependent synaptic plasticity was also found in the central auditory system, as in vivo LTP was found to be saturated at DCN synapses after 4–5 days of sound exposure that led to tinnitus-like symptoms in an animal model, which was later reversed using an NMDA receptor antagonist [177]. Studies from our laboratory also found that in vivo LTP was facilitated in the inferior colliculus of rats at five months after acoustic trauma and the extent of LTP facilitation was similar to that observed following the administration of a GABA_A_ receptor antagonist picrotoxin in normal rats (unpublished observations). This suggests that acoustic trauma may cause long-term enhancement of synaptic plasticity in the inferior colliculus, which may be due to a loss of inhibition similar to that observed following the inhibition of GABA_A_ receptors. Therefore, there seems to be a link between noise exposure and/or tinnitus and altered synaptic plasticity in the auditory pathways.

However, there is no evidence directly connecting tinnitus to changes in synaptic plasticity in the limbic system, given that auditory gating in the CA3 region of the rat hippocampus is disrupted following LTP stimulation (three trains of 250 Hz/1 s stimulation) [178]. One study has linked the alteration of theta rhythm in the hippocampus to the impairment of auditory gating mechanisms [179], however, more research is required to look into the mechanisms contributing to alteration in theta rhythms. Theta rhythm is widely recognized to have a role in the mechanisms of synaptic plasticity in the hippocampus [180]. Given that an impaired auditory gating function is proposed to be involved in tinnitus and changes in synaptic plasticity could disrupt auditory gating in the hippocampus, it is conceivable that synaptic plasticity in the limbic areas would also be altered in tinnitus. Studies have demonstrated the link between noise exposure and hippocampal plasticity. For example, noise exposure has been shown to cause changes in the intrinsic membrane properties of hippocampal pyramidal cells [181], resulting in impaired plasticity and reduced phosphorylation of plasticity-related signalling molecules [182]. Intense noise exposure also caused granule and pyramidal cell dysfunction [183] and significantly altered place cell activity and hence hippocampal plasticity [184] (Figure 2). Moreover, long-term exposure to high-intensity sound potentiated the amplitude of the inhibitory GABAergic currents, indicating that high-intensity sound exposure may impair hippocampal inhibitory transmission and, as a result, alter synaptic plasticity [185]. Indeed, LTP in the hippocampus was found to be impaired following a single episode of the high-intensity sound exposure [186]. Another study reported that noise exposure impairs hippocampal-mediated learning and memory functions by reducing LTP induction and downregulating some important LTP signalling molecules such as Ca^2+^/calmodulin-dependent protein kinase (CaMKII) at the hippocampal synapses [182].

In addition to the hippocampus, fos-like immunoreactivity was significantly increased in the amygdala of animals that were exposed to intense noise causing tinnitus [187]. The amygdala nuclei respond substantially to sound (traumatic and non-traumatic) immediately, and over one month after sound exposure [187], which suggests that the amygdala may undergo long-term plasticity in response to noise and/or tinnitus. Therefore, further investigation is required to link tinnitus with changes in synaptic plasticity in the limbic areas.

## 4. Potential Molecular/Receptor Targets Mediating Tinnitus

Although there is ample evidence to suggest that the limbic system may be involved in tinnitus through altered synaptic plasticity, the underlying molecular mechanisms remain elusive, for example, the receptor subtypes and their associated downstream signalling that produces the effects.

One potential target could be the endocannabinoids and their receptor subtypes, i.e., the CB_1_ receptor (cannabinoid receptor subtype 1) which is found primarily in the brain and acts as a neuromodulatory receptor, and the CB_2_ receptor (cannabinoid receptor subtype 2) which is found primarily in immune-derived cells and produces anti-inflammatory effects [188]. In the auditory cortex, neocortex, hippocampus, basal ganglia, cerebellum, and brainstem, the CB_1_ receptor is present at high levels [189,190] and plays a critical role in mediating synaptic plasticity [191]. The endocannabinoids in the auditory pathways have been demonstrated to affect glutaminergic [192], glycinergic [193] and cholinergic [194] signals, thereby modulating auditory function. For example, in the cochlear nucleus, CB_1_ receptors mediate depolarization-induced suppression of inhibition and excitation [193], and long-term depression [195], indicating the involvement of the endocannabinoid system in the modulation of plasticity in the area [196]. In rats, agonists at the CB_1_ receptor impaired auditory gating [197,198] and resulted in the impairment of hippocampal LTP [199] and LTD [200] mechanisms. Studies have shown that long-term exposure to delta-9-tetrahydrocannabinol (Δ-9-THC), the main psychoactive component of *Cannabis*, reduced input from the mPFC to the NAcc while increasing the input from the ventral hippocampus and basolateral amygdala to the NAcc [201] (Figure 3). The CB_1_ receptor was significantly down-regulated in the ventral region of the cochlear nucleus in a salicylate-induced animal model of tinnitus [202], and an agonist at the CB_1_ receptor was found to promote the development of tinnitus following the administration of salicylate [203]. However, CB_1_ receptor activation has also been reported to suppress anxiety responses and amygdala reactivity to unpleasant stimuli [204] which could be due to its role in modifying neuronal activity in the basolateral amygdala upon activation [205]. Taken together, endocannabinoids may play an important but complicated role in mediating tinnitus through their modulatory effects on synaptic plasticity and more research into the pathophysiological significance of the endocannabinoid systems in auditory and limbic areas using animal models of tinnitus may open doors for better understanding of tinnitus.

In addition to the above, dopamine, a principal neurotransmitter of the basal ganglia known to play a critical role in Parkinson’s Disease, motor control, psychological activity and dependence [206,207,208], may also play a role in mediating tinnitus [209,210]. Mostly, the dopaminergic neurons are found in the ventral tegmental region and substantia nigra [211], and may have a role in influencing the auditory pathway’s afferent neurotransmission between the limbic system and cochlea [212], indicating some networking between the two regions. Dopaminergic pathways have been linked with tinnitus and its management, as it is proposed to support the functional neuroanatomy of tinnitus perception. For example, in a human trial involving 120 patients, sulpiride (a selective antagonist of dopamine D_2_, D_3_ and serotonin 1A receptors) decreased tinnitus perception by 56%, melatonin (a free radical scavenger) administration decreased it by 40%, and sulpiride along with melatonin decreased it by 81% [213]. One reason behind the low success rate of clinical trials relating to tinnitus could be due to the lack of identifying the primary target and its manipulation through different drug dosages. Thus, more investigation is needed to better understand the potential efficacy of dopaminergic drugs. Dopaminergic neurons in the ventral tegmental area (VTA) send projections to the limbic areas, and dopamine has been implicated in modulating auditory sensory gating in both human [214] and animal [215,216,217] studies and is an essential element in the brain reward system [218]. Given that direct injection of a dopamine agonist into the NAcc significantly decreased auditory gating in the hippocampus [217], it is conceivable that dopamine neurotransmission in the limbic system may be crucial in keeping the auditory gating functional and its dysfunction may bring tinnitus to conscious perception. Therefore, more studies for a better understanding of dopamine involvement in tinnitus are needed.

Finally, most of the previous research has focused on neuronal changes in the brain, but glial cells may be important as well. For example, noise-induced hearing loss causes increased production of proinflammatory cytokines and microglial activation in the primary auditory cortex, indicating neuroinflammation. Cytokines such as the tumour necrosis factor (TNF) are also released from the astrocytes and microglia [219]. In TNF knockout mice or during blockade of TNF expression pharmacologically, neuroinflammation is reduced and the behavioural phenotype associated with tinnitus in animals is improved [220]. Furthermore, acoustic damage can elicit proinflammatory cytokines (TNF and interleukins) in glial cells such as astrocytes and microglial cells [221], which are now well-recognised as passive mediators of synaptic plasticity and neurotransmission [166]. Thus, investigating the role of the proinflammatory molecules and glial cells in the auditory and limbic system plasticity following noise trauma could yield unique information.

## 5. Effective Strategies and Future Directions

Despite using various drugs and other treatment strategies, effective treatments for tinnitus are still lacking. This is probably because there is no effective approach to target auditory and non-auditory (such as the limbic system) systems at the same time. Thus, addressing the system dynamics in tinnitus pathogenesis, especially the limbic component of tinnitus persistence, will facilitate the development of a more comprehensive combination of pharmaceutical therapies.

Furthermore, tinnitus is frequently linked with psychological stress, and certain types of stress are known modifiers of epigenetic markers in both humans and animals, [222,223] which indicates that tinnitus may also emerge from epigenetic modifications. Many studies have linked these epigenetic modifications to experimentally generated behavioural changes that have also been reported in patients suffering from depression or anxiety [224]. It is also reported that the pattern of epigenetic changes differs in healthy versus hearing-impaired patients [225,226]. Since tinnitus sufferers have a significantly higher rate of hearing loss, it is reasonable to think that at least some epigenetic targets are shared between the two disorders.

## 6. Conclusions

Although there is now mounting evidence from human neuroimaging studies for tinnitus-related changes in both auditory and limbic system areas, the findings are limited, and we are yet to develop a tool that can address converging or overlapping biological pathways and distinct brain structural alterations associated with chronic tinnitus. Similarly, our understanding of the biochemical underpinnings of tinnitus such as the molecular, cellular, and system-level mechanisms is limited. The literature suggests that the limbic system and adjacent areas may be involved in tinnitus development and hence these regions may be critical for tinnitus treatment as well. Therefore, future tinnitus interventions will benefit from a better knowledge of auditory–limbic interactions.

## Figures and Tables

**Figure 1 ijms-24-09889-f001:**
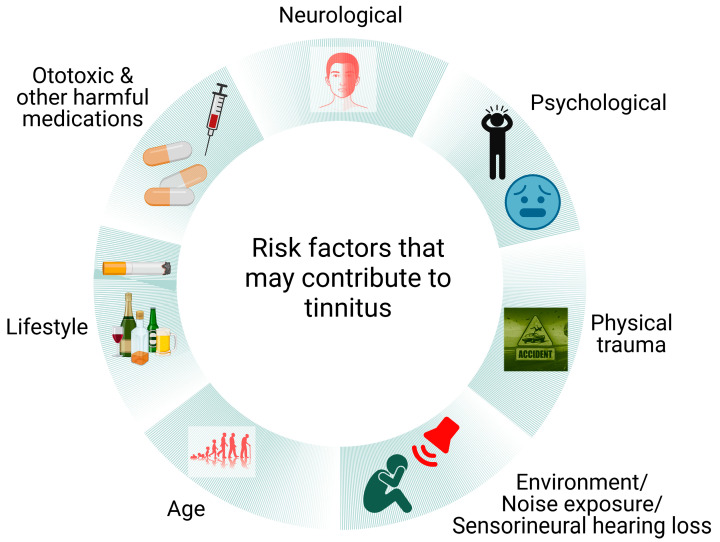
**Risk factors contributing to tinnitus.** Tinnitus induction is heterogeneous in nature as the onset can occur due to neurological issues (e.g., migraine, epilepsy), psychological issues (e.g., depression, anxiety), physical trauma, excessive noise exposure, sensorineural hearing loss, presbycusis (age-related), lifestyle (e.g., excess smoke and alcohol consumption) and ototoxic medications.

**Figure 2 ijms-24-09889-f002:**
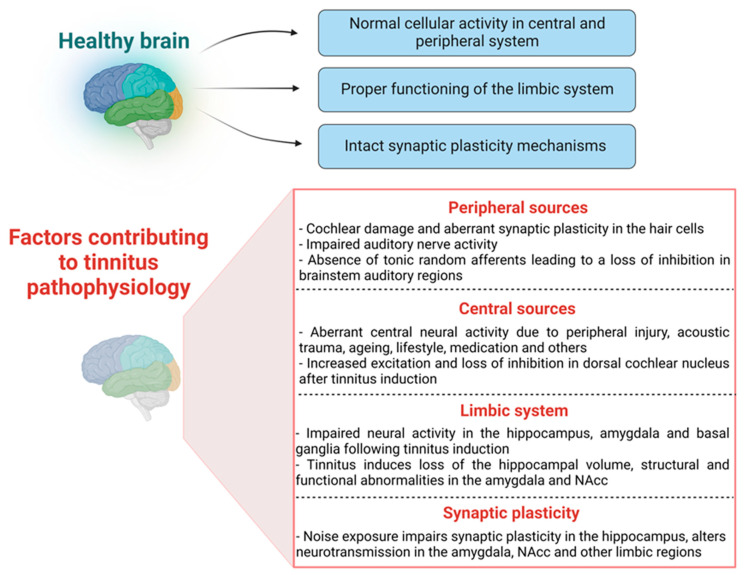
Comparison of differential regulation of neural activity, synaptic transmission, and plasticity at different peripheral and central levels in healthy versus tinnitus brain. (NAcc- nucleus accumbens).

**Figure 3 ijms-24-09889-f003:**
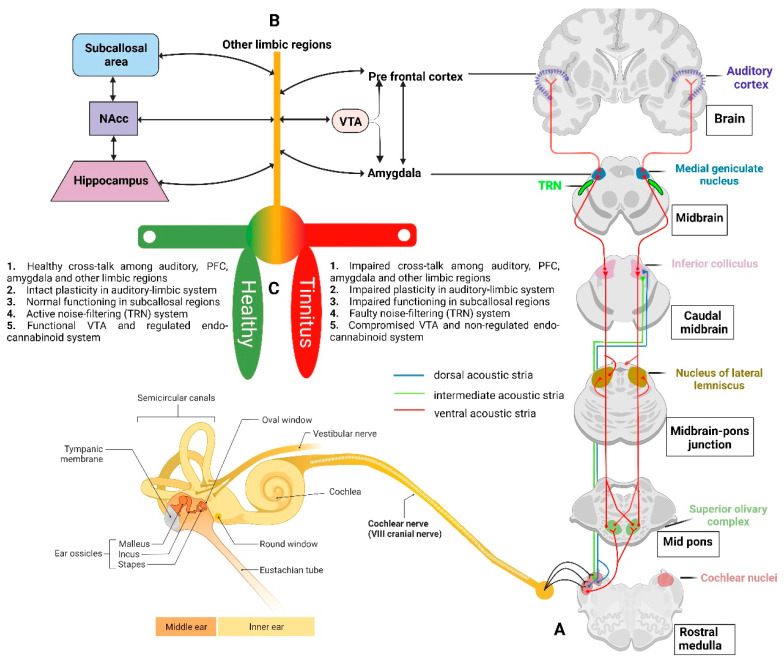
(**A**) Schematic figure of the auditory pathway from the cochlea to the auditory cortex and its interaction with the limbic system. Sound-evoked information is relayed from the cochlea via the cochlear nerve (8th cranial nerve) to the cochlear nucleus present in the rostral medulla. The cochlear nucleus sends neuronal projections to higher brain regions through three main pathways-the dorsal, intermediate, and ventral acoustic stria. The ventral stria sends projections to the superior olivary complex. Together with the axons from the cochlear nuclei, postsynaptic axons from the superior olivary nucleus project to the inferior colliculus in the midbrain. The medial geniculate nucleus of the thalamus receives axons from cells in the colliculus and projects its axons to the primary auditory cortex for conscious perception. (**B**) The limbic areas such as the amygdala and pre-frontal cortices receive the signal for the evaluation of sound content. It has been proposed that on the arrival of unpleasant sound frequencies, the medial geniculate nucleus is inhibited by the thalamic reticular nucleus (TRN) and thus acts as a noise-filtering system. Furthermore, the ventral tegmental area (VTA) is also connected via the endocannabinoid system to the pre-frontal cortex and amygdala which sends and receives information from other limbic regions such as the hippocampus, nucleus accumbens (NAcc), subcallosal areas and others. (**C**) The outstanding factors that contribute to a healthy or tinnitus brain are the interactions and plasticity in the limbic and subcallosal area, TRN noise-filtering system and cannabinoid system.

## Data Availability

The data used in this review article are obtained from publicly available literature, published articles, and clinical trials. All sources and references are appropriately cited within the article. No datasets were generated or examined for this review, and therefore, no additional data are available beyond what is provided in the manuscript.

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
