# Peer review of "Targeting the Limbic System: Insights into Its Involvement in Tinnitus"

_ijms, 2023, doi:10.3390/ijms24129889_

Round 1

Reviewer 1 Report

This is an excellent review, and it is very well written. I recommend its publication. I only have 3 comments for the authors.

1. The distinction between tinnitus and tinnitus disorder is relevant to explain the role of the limbic system in tinnitus development.

2. There is no mention to the heritability and genetic contribution to tinnitus and this should be mentioned in the introduction and in the hippocampus and tinnitus section. Rare variants in several human genes have demonstrated to be associated with severe tinnitus, including ANK2 or CACNA1E. Although functional studies on synaptic function are needed, these genes are highly expressed in the hippocampus and cortex in rodents suggesting an increase susceptibility to develop severe tinnitus. 

3. In figure 1, sensorineural hearing loss should be included since it is the most important risk factor.

Suggested references

Amanat S, Gallego-Martinez A, Sollini J, Perez-Carpena P, Espinosa-Sanchez JM, Aran I, Soto-Varela A, Batuecas-Caletrio A, Canlon B, May P, Cederroth CR, Lopez-Escamez JA. Burden of rare variants in synaptic genes in patients with severe tinnitus: An exome based extreme phenotype study. EBioMedicine. 2021 Apr;66:103309. doi: 10.1016/j.ebiom.2021.103309. Epub 2021 Apr 1. PMID: 33813136; PMCID: PMC8047463.

Gallego-Martinez A, Escalera-Balsera A, Trpchevska N, Robles-Bolivar P, Roman-Naranjo P, Frejo L, Perez-Carpena P, Bulla J, Gallus S, Canlon B, Cederroth CR, Lopez-Escamez JA. Using coding and non-coding rare variants to target candidate genes in patients with severe tinnitus. NPJ Genom Med. 2022 Nov 30;7(1):70. doi: 10.1038/s41525-022-00341-w. PMID: 36450758; PMCID: PMC9712652.

Author Response

Comment 1. The distinction between tinnitus and tinnitus disorder is relevant to explain the role of the limbic system in tinnitus development.

Reply: We have added a few sentences as below-

Introduction, Pg1, line 25. Tinnitus and tinnitus disorder are the two main classifications used to describe tinnitus and tinnitus-related distress. While tinnitus is the conscious perception of a tone or composite noise that has no corresponding external auditory source, tinnitus disorder can include cognitive impairment, emotional discomfort and functional disability [1].

Comment 2. There is no mention to the heritability and genetic contribution to tinnitus and this should be mentioned in the introduction and in the hippocampus and tinnitus section. Rare variants in several human genes have demonstrated to be associated with severe tinnitus, including ANK2 or CACNA1E. Although functional studies on synaptic function are needed, these genes are highly expressed in the hippocampus and cortex in rodents suggesting an increased susceptibility to develop severe tinnitus. 

Reply: We have added a few sentences and inserted the references as suggested by the reviewer in the manuscript as follows-

1. Introduction, pg1, line 34. Due to the inclusion of people with self-reported tinnitus and high phenotypic variation in large genomic studies, the extent of genetic involvement in severe tinnitus is still unknown[11]. However recent studies have identified rare genes that are associated with severe tinnitus [12-13].

2.2. The hippocampus and tinnitus, Pg 4, line 143.  Genetic factors have recently been associated with severe tinnitus, such as an increased occurrence of rare variants in ANK2 and TSC2 synaptic genes, which exhibit high expression in the hippocampus and cortex[12]. Furthermore, one study reported an increased occurrence of specific genetic changes such as missense and large structural variants in regions of the genome that are highly conserved.  They identified CACNA1E, NAV2, and TMEM132D as potential genes that may play a role in contributing to severe tinnitus[13]. However, additional research is necessary to establish a definitive genetic linkage between tinnitus and the hippocampus.

Comment 3. In Figure 1, sensorineural hearing loss should be included since it is the most important risk factor.

Reply: Sensorineural hearing loss is now added along with environment, and noise exposure section in Fig. 1, pg 10.

Please note that we have highlighted the changes in the attached manuscript.

Reviewer 2 Report

The study titled: "The role of the limbic system in tinnitus development: Insights for potential target-specific therapies", is a very interesting review paper that aims to compile all the existing information on its etiopathology, pharmacology and the pathways that support it and keep it in the consciousness of patients.

The study is well-structured and written, it is extensive and presents some figures that help its reading. However, I would like to suggest to the authors that because the title of the study is "... Insights for potential target-specific therapies". The authors could include, either in Figure 3 or in a new one, the anatomical and molecular targets of future and present therapies for its control, as well as highlight the importance of the studies on which they are based.

Author Response

Reply: We would like to express our gratitude to the reviewer for providing valuable feedback on our manuscript. We appreciate the reviewer's recognition of the molecular targets relevant to current and future therapies in tinnitus. However, as authors, our primary focus in this manuscript was to investigate the involvement of limbic regions in tinnitus. “Target-specific therapies” term was used in the title to reflect the limbic regions as the primary target in the condition that may be used in future as therapeutics. Consequently, we intentionally chose not to delve into extensive detail regarding the molecular targets associated with tinnitus, and now we have tweaked our title to keep it consistent with our manuscript as suggested by the reviewer-

Targeting the limbic system: insights into its involvement in tinnitus.

We acknowledge the importance of these targets and their potential relevance in treatment approaches, and we will ensure to address this aspect more explicitly in future studies or related publications. Thank you again for your insightful comments.

Please be advised that we have made a slight modification to the manuscript, and it has been highlighted for your convenience.
